
# Statistical characteristics of raindrop size distribution during rainy seasons in Complicated Mountain Terrain

Wenqian Mao[1,2,3], Wenyu Zhang[1,2,3*], Menggang Kou[1]

1. School of Geoscience and Technology, Zhengzhou University, Zhengzhou, 450001, China

2. Key Laboratory for Cloud Physics, Chinese Academy of Meteorological Sciences, Beijing 100081, China

3. College of Atmospheric Sciences, Lanzhou University, Lanzhou, 730000, China

*Correspondence to*: Wenyu Zhang (zhangwy@zzu.edu.cn)

**Abstract**:In order to understand the differences of raindrop size distribution (DSD) in complex mountainous terrain, the characteristics of DSD were analyzed by using the six-months observation data at the southern slopes, northern slopes and inside in Qilian Mountains. For all rainfall events, the number concentration of small and large raindrops on the inside and south slope are greater than that on the north slope, but midsize raindrops are less. The DSD spectrum of inside mountains are more variable and significantly differ from the north slopes. The differences in normalized intercept parameters of DSD for stratiform and convective rainfall are 8.3% and 10.4%, respectively, and mass-weighted diameters are 10.0% and 23.4%, respectively, which the standard deviation of DSD parameters on inside sites are larger. The differences in coefficient and exponent of Z-R relationship are 2.5% and 10.7%, respectively, with an increasing value of coefficient from the south slope to the north slope in stratiform rainfall but opposite to convective rainfall. In addition, the DSD characteristics and Z-R relationships are more similar at the ipsilateral sites and have smaller differences between the south slope and inside mountains.

**Keywords**: *Raindrop size distribution; Complicated mountain terrain;* characteristic difference



## 1 Introduction

Raindrop size distribution (DSD), the number of raindrops per drop size per unit volume, is an important parameter to statistically describe the microstructure of precipitation(Bringi et al., 2003; Ma et al., 2019a). The measurement of DSD can provide some fundamental information such as raindrop size (D), liquid water content (W), rain rate (R), radar reflectivity factor (Z) and so on, which has an essential contribution to improving quantitative precipitation estimates (QPE) using weather radar and satellite observations (Adirosi et al., 2018; Jash et al., 2019). The parameterization of DSD can obtain the distribution model parameters of DSD in different rain types, which is significant to advance microphysics parameterization in numerical weather prediction (NWP) models (Wainwright et al., 2014; McFarquhar et al., 2015; Zhao et al., 2019). In addition, understanding the DSD is crucial in many application fields concerning hydrology, agriculture, soil erosion and microwave communication (Rincon et al., 2002; Smith et al., 2009; Angulo-Martínez et al., 2015; Lim et al., 2015; Yang et al., 2016).

Numerous studies have been carried out the statistical characteristics of DSD in different regions (Campos et al., 2006; Seela et al., 2017; Dolan et al., 2018; Protat et al., 2019; Loh et al., 2019; Jash et al., 2019). It is shown that the number concentration and size of raindrops increase with rain rate and so DSD becomes higher and wider. The characteristics in different rain types display that the mass-weighted mean diameter (i.e., $D_m$) and normalized intercept parameter (i.e., $N_w$) of convective rainfall (CR) are larger than those of stratiform rainfall. (SR). Furthermore, these studies also reveal that there are more differences in characteristics of DSD. Dolan et al. (2018) divided global DSD characteristics into 6 types by using 12 datasets across three latitudes and found centralized regions and DSD parameters of 6 types varied in location. The average number of raindrops in central Korea were usually more numerous than that in southeast under three rainfall systems, especially drops on 0.31-0.81mm diameter range (Loh et al., 2019). According to the DSD results from Tibetan Plateau (TP), it showed the eastern regions had higher number concentration of raindrops on 0.437-1.625mm diameters and more variation on different diameters than that in central regions (Wang et al., 2020). Compared to eastern China and northern China, the DSD in southern China demonstrated a higher number concentration of relatively small-sized drops, respectively (Zhang et al., 2019). The comparison of Z-R relationship (defined as $Z=AR^b$) indicated that coefficient decreased with increasing R in southern TP, which is opposite in south China (Wu et al., 2017). For the DSD parameters of SR and CR, there are various changes between the lower reaches and middle reaches of Yangtze River (Fu et al., 2020).

As reported in above studies, DSD characteristics significantly vary with factors such as geographical location, climatic region and rain types. Pu et al. (2020) analyzed the DSD characteristics of five sites in Najing city and found $N_w$ of DSD was largest at site near industrial areas but $D_m$ of DSD was largest at site near city's centre. In other words, even at urban scale, there are still differences to microphysical characteristics reflected by the DSD which is due to the influence of the surrounding environment. Then how do the characteristics of DSD vary from location for the complicated





mountain terrain? Rao et al. (2006) suggested that the obvious variation in DSD with
altitude were related to evaporation and breakup by comparing the DSD parameters at
different altitudes. Geoffroy et al. (2014) concluded that the total concentration of
raindrops decreased while the average drop size increased as decreasing altitude, which
used aircraft observations. Then how large would be the differences in DSD at different
altitudes in mountainous region? And then how significant would be the effects of these
differences?
Qilian mountains, a series of marginal mountains in the northeastern part of TP,
are the vitally important ecological protection barrier in northwest arid areas, which
block the connection of deserts and wilderness in the northwest (shown as Figure 1).
The mountains form several inland rivers that are important water source for the
northwest arid areas and have made a considerable contribution to regional economic
development (Gou et al., 2005; Tian et al., 2014; Qin et al., 2016). In this paper, we
choose Qilian mountains as the research object and select 6 sites with different
backgrounds representing the southern slopes, northern slopes and inside of Qilian
mountains. To thoroughly investigate the discrepancies in the complicated mountain
terrain, the DSD characteristics and Z-R relationships are comprehensively analyzed
according to different rain types based on continuous disdrometer observations in rainy
season. The primary goal is to obtain the finer precipitation of Qilian mountains and
improve the accuracy of QPE, which would be as research foundation for developing
cloud water resources in mountainous areas.
**2  Data and method**
**2.1 Sites and instruments**
The eastern and middle sections of the Qilian Mountains were chosen as the main
study area, taking into account that several important inland rivers originating from
these areas of Qilian Mountains (Li et al., 2019). Six disdrometers were deployed on
the southern slopes, northern slopes and inside (close to the ridge) of Qilian mountains,
with three sites in the eastern section which called Taola (TL), Huangchengshuiguan
(HS), and Liuba (LB) from south to north, and with another three sites in the middle
section which called Daladong (DLD), Boligou (BLG), Shandan (SD) from south to
north. The background of Qilian Mountains is shown on the satellite map, and the six
sites are marked on the topographical map, as Figure 1. The distances between six sites
are listed in Table 1. The sites on the south, north and inside are basically parallel to the
trend of the mountain, and the sections formed by the sites in the east and middle are
basically perpendicular to the trend of the mountain. Through the historical weather
review and rain gauge observation results, the rainy season at the six sites is
concentrated in May to October, with more precipitation in July, August and September.






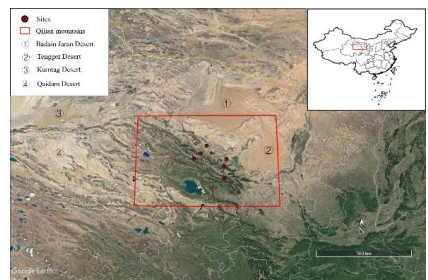


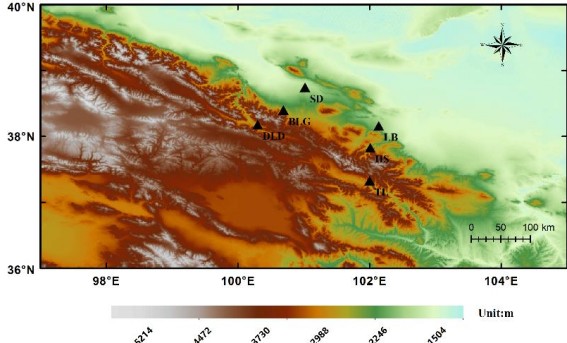

Fig.1 The Geographical overview of Qian mountains and the disdrometer sites; the
circles or triangles represent the location of the sites. The map above is from Google
Earth © Google Earth
Table 1 Location between every two sites (latitude, longitude, sea level height and
distance information).

| Six sites distance (km) | LB | HS | TL | SD | BLG | DLD |
|---|---|---|---|---|---|---|
| LB (38.16˚N, 102.14˚E, 1926m) | - | 39.6 | 94.3 | 116.0 | 129.6 | 161.1 |
| HS (37.83˚N, 102.01˚E, 2342m) | - | - | 55.6 | 135.1 | 132.8 | 154.9 |
| TL (37.33˚N, 102.00˚E, 2910m) | - | - | - | 182.4 | 167.3 | 177.0 |
| SD (38.80˚N, 101.08˚E, 1765m) | - | - | - | - | 54.2 | 96.8 |
| BLG (38.4˚N, 100.69˚E, 2455m) | - | - | - | - | - | 43.3 |
| DLD (38.18˚N, 100.3˚E, 2957m) | - | - | - | - | - | - |

This experiment used an optical, laser-based device to measure DSD, called DSG4
disdrometer, which met the assessment of Functional Specification Requirements For
Disdrometer issued by the China Meteorological Administration. The disdrometer has
the HSC-OTT Parsivel2 sensor as observation part manufactured by OTT Messtechnik
(Germany) and Huatron (China). When raindrops pass through the horizontal flat laser
beam generated by the transmitting part of the instrument, it causes the signal
attenuation in the laser observation area. The raindrop size is determined by the degree
of signal attenuation and the falling speed is recorded by the transit time. The sampling
time is 60s and the velocity and drop sizes are divided into 32 non- equally spaced bins,
varying from 0.05 to 20.8 m s−1 for velocity and 0.062 to 24.5 mm for drop diameter.





**2.2 Quality control of the data**
It is necessary to carry out quality control on the data due to potential instrument
error. Every minute of DSD has been carefully processed, which collected by the six
DSG4 disdrometers from May to October 2020. The following criteria have been
employed in choosing data for analysis (Zhang et al., 2019). (1) The first two size bins
were ignored because of low signal-to-noise ratio; (2) samples with 1-min total number
of raindrops less than 10 or rain rate at moment of discontinuous observation less than
0.1 mmh$^{-1}$ were regarded as noise; (3) raindrops at the diameter of more than 8 mm
were eliminated; (4) raindrop with a falling terminal velocity $V(D_i)$ that deviates from
the empirical terminal velocity $V_{emp}(D_i)$ more than 40% were removed (Kruger and
Krajewski, 2002); and (5) samples with less than 5 bins after the correction of falling
terminal velocity were deleted because its DSD can't be determined with too few bins.
$$\left|V(D_i) - V_{emp}(D_i)\right| < 0.4V_{emp}(D_i) \tag{1}$$
where $V_{emp}(D_i) = 9.65 - 10.3\exp(-0.6D_i)$ ($D_i$ is the mean volume-equivalent
diameter of the ith size category), as derived from the formula given in Atlas et al.
142 (1973).

**2.3 Integral parameters of rainfall**
The basic observations obtained by disdrometer is counts of raindrops at each
diameter and velocity. And the diameters given by disdrometer are the mid value of two
adjacent bins, which we take the diameters as the corresponding endpoint bin values.
The velocities are weighted average velocity class over the corresponding disdrometer.
The raindrop number concentration $N(D_i)$ (m$^{-3}$mm$^{-1}$) in the ith size bin per unit volume
per unit size interval for diameter is calculated the following equation:
$$N(D_i) = \sum_{i,j=1}^{32} \frac{n_{i,j}}{A \cdot \Delta t \cdot V_j \cdot \Delta D_i} \tag{2}$$
Where $n_{i,j}$ is the counts of raindrops measured by disdrometer within the size bin i and
velocity bin j during sampling time $\Delta t$; A and $\Delta t$ are the sampling area (0.0054 m$^2$) and
sampling time (60 s), respectively; $V_j$ (m s$^{-1}$) is the mid-value falling speed for velocity
bin j; $\Delta D_i$ is the diameter spread for the ith diameter bin.
Some integral rainfall parameters, such as total number concentration $N_t$ (m$^{-3}$), rain
rate R (mm h$^{-1}$), radar reflectivity factor Z (mm$^6$ m$^{-3}$) and liquid water content W (g
cm$^{-3}$), can be derived by the following equation:
$$N_t = \sum_{i=1}^{32} N(D_i)\Delta D \tag{3}$$
$$R = \frac{6\pi}{10^4 \rho_w} \sum_{i=1}^{32} V(D_i)\, D_i^3 N(D_i)\Delta D_j \tag{4}$$
$$Z = \sum_{i=1}^{32} N(D_i)D_i^6 \Delta D_i \tag{5}$$



$$W = \frac{\pi \rho_w}{6 \times 10^3} \sum_{i=1}^{32} D_i^3 N(D_i) \Delta D_i \qquad (6)$$
where $\rho_w$ is water density (1.0 gcm$^{-3}$); $V(D_i)$ is the falling speed measurements from
disdrometer. In this study, when calculating rain rate we use $V_{emp}(D_i)$ to replace $V(D_i)$
because of measurement error, particularly at larger bins and faster falling speeds.
The DSD characteristics can be described by three-parameter gamma distribution
in following form. And it has better capability than M-P distribution to describe the
broader variation of DSD fluctuations, which has been proven to be well fitted the main
part of spectra and reduce the fitting error on small and large scale.
$$N(D) = N_0 D^\mu \exp(-\Lambda D) \qquad (7)$$
where $N(D)$ is the raindrop number concentration; D is the raindrop bins with unit mm;
$N_0$, $\mu$ and $\Lambda$ are intercept, shape and slope parameter from three parameters of gamma
model which can be derived from gamma moments or least square method, respectively.
When $\mu=0$, it degenerates into M-P DSD model.
Although, three-parameter gamma distribution is commonly accepted model, the
normalized gamma model has been widely adopted with its independent parameters
and clear physical meaning as follows:
$$N(D) = \frac{3}{128} N_w \left[ \frac{(4+\mu)^{(4+\mu)}}{\Gamma(4+\mu)} \right] \left( \frac{D}{D_m} \right)^\mu \exp\left( \frac{-(4+\mu)D}{D_m} \right) \qquad (8)$$
Where $\mu$ is the shape parameter in dimensionless; $D_m$ (mm) is the mass-weighted mean
diameter and $N_w$ (m$^{-3}$ mm$^{-1}$) is the normalized intercept parameter computed from $D_m$.
The form is as follows:
$$D_m = \frac{\sum_{i=1}^{32} N(D_i) D_i^4 \Delta D_i}{\sum_{i=1}^{32} N(D_i) D_i^3 \Delta D_i} \qquad (9)$$
$$N_w = \frac{4^4}{\pi \rho_w} \left( \frac{10^3 W}{D_m^4} \right) \qquad (10)$$
**3   DSD parameter characteristics**
**3.1 Characteristics of DSD**
The number of 1 min DSD spectra from six sites have been selected after data
quality control covering the rainy season (May-October) in the Qilian Mountains region
in 2020, which are accounted for 87.9%, 85.8%, 84.5%, 91.2%, 80.6%, 86.5% of the
total number of samples to LB, HS, TL, SD, BLG, DLD, respectively. Figure 2a shows
the mean DSDs for the six districts in Qilian mountains. The maximum concentration
of raindrops is around on 0.562mm diameter and the maximum number concentration
values of sites are BLG>TL>DLD>HS>SD>LB. As the increasing diameter, the
number concentration values decrease and the concentration values are
LB>SD>DLD>TL>BLG>HS at around 2 mm diameter. When the diameter is larger
than 4 mm, the concentration of TL, BLG and HS are relatively high. In this study, it is
roughly divided into small raindrops (less than 1 mm in diameter), midsize raindrops



(1-3 mm) and large raindrops (greater than 3 mm) to easily describe the difference of
DSDs (Ma et al., 2019b; Pu et al., 2020). To highlight the DSD differences caused by
background environment, Figure 2b shows the mean DSDs normalized with $N_w$ and $D_m$
results for sites. Compared with Figure 2a, the characteristics of the raindrops are more
consistent across sizes, while the differences between the above sites are more
pronounced, especially in the medium and large raindrops, which truly reflects the DSD
differences caused by location variability. Combining the characteristics of the
geographical environment of the six sites, we can analyze some differences in DSD
characteristics in Qilian Mountains. For small raindrops, the number concentrations on
the inside and southern slopes districts are greater than that on the northern slopes; for
midsize raindrops, the number concentrations decrease sequentially on the northern
slopes, southern slopes and inside districts; for large raindrops, the number
concentrations on the inside districts are larger. In addition, the number concentrations
of raindrops in the middle section of the mountainous area is slightly greater than that
in the eastern section.

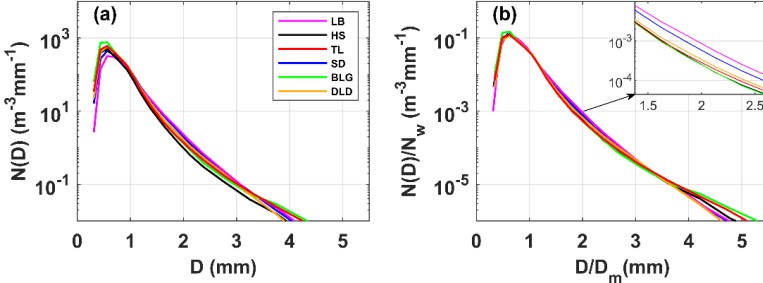

Fig.2 (a) Mean measured DSDs; (b) Normalized mean DSDs at six sites of Qilian
mountains region in rainy season
**3.2 Distribution of DSD parameters**

In order to study the differences in DSDs, we selected 6 integral rainfall parameters
for discussion, which are normalized intercept parameter ($N_w$), mass-weighted mean
diameter ($D_m$), shape parameter ($\mu$), total number concentration ($N_t$), rain rate (R) and
radar reflectivity factor (Z). Figure 3 and Table 2 show the distribution and statistics of
6 DSD parameters (the distribution of each parameter is normalized using the uniform
method). Averagely, Dm is more concentrated on smaller values at HS and BLG, which
shows smaller mean values than TL and DLD, while significantly more values greater
than 1mm at LB and SD; log10Nw is more centralized on larger values at TL and DLD,
with relatively smaller values at LB and SD; the distribution patterns for $\mu$ and log10Nt
are similar to those for log10Nw. The density curves of R and Z are similar, but there
are differences at the 6 sites, which would be analyzed in detail in subsequent content.
It is noteworthy that the frequency of samples with R around 0.6-1.0 mmh[-1] is highest,
and samples with R less than 1mmh[-1] account for more than half of the total rainfall.



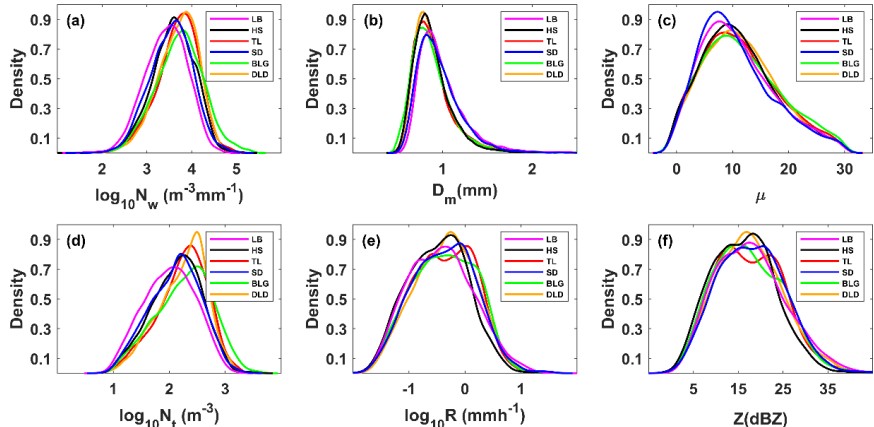


Fig.3 Probability density distribution of integral DSD parameters at six sites (LB, HS,
TL, SD, BLG, DLD): (a) normalized intercept parameter log10Nw (m$^{-3}$mm$^{-1}$); (b)
mass-weighted mean diameter Dm (mm); (c) shape parameter μ; (d) total number
concentration $\log_{10}N_t$ (m$^{-3}$); (e) rain rate R (mmh$^{-1}$); (f) radar reflectivity factor Z
(mm$^6$mm$^{-3}$)
Table 2 Statistical of several integral DSD parameters for all observations at six sites
(LB, HS, TL, SD, BLG, DLD).

| Sites | $\text{Log}_{10}N_w$ (m$^{-3}$mm$^{-1}$) | | | $D_m$ (mm) | | | μ | | | $\text{Log}_{10}N_t$ (m$^{-3}$) | | | R (mmh$^{-1}$) | | | Z dBZ | | |
|---|---|---|---|---|---|---|---|---|---|---|---|---|---|---|---|---|---|---|
| | ME | SD | SK | ME | SD | SK | ME | SD | SK | ME | SD | SK | ME | SD | SK | ME | SD | SK |
| LB | 3.43 | 0.47 | -0.25 | 0.99 | 0.29 | 2.68 | 10.92 | 6.63 | 0.61 | 2.01 | 0.46 | -0.07 | 0.94 | 1.90 | 0.23 | 17.79 | 7.82 | 0.44 |
| HS | 3.59 | 0.48 | -0.29 | 0.89 | 0.25 | 3.35 | 11.12 | 6.64 | 0.53 | 2.13 | 0.45 | -0.22 | 0.69 | 1.60 | 0.05 | 16.24 | 7.08 | 0.34 |
| TL | 3.69 | 0.48 | -0.55 | 0.90 | 0.29 | 4.49 | 11.37 | 6.84 | 0.48 | 2.23 | 0.44 | -0.43 | 0.89 | 1.48 | -0.05 | 17.47 | 7.55 | 0.35 |
| SD | 3.54 | 0.48 | -0.17 | 0.96 | 0.26 | 2.12 | 10.62 | 6.61 | 0.71 | 2.11 | 0.46 | -0.17 | 0.97 | 2.01 | 0.06 | 17.95 | 7.47 | 0.28 |
| BLG | 3.72 | 0.54 | -0.15 | 0.89 | 0.29 | 5.17 | 11.71 | 7.06 | 0.46 | 2.26 | 0.50 | -0.25 | 0.94 | 2.13 | -0.04 | 17.34 | 7.66 | 0.41 |
| DLD | 3.69 | 0.45 | -0.50 | 0.90 | 0.25 | 2.66 | 11.52 | 6.66 | 0.43 | 2.24 | 0.43 | -0.46 | 0.95 | 1.62 | -0.01 | 17.70 | 7.43 | 0.37 |

Note: ME is mean; SD is standard deviation; SK is skewness.
**3.3 Characteristics of DSD in different rain rate classes**
To further understand the characteristics of DSDs at the six sites, the samples are
divided into six classes according to the associated rain rate (R): C1, R<0.5; C2,
0.5≤R<2; C3, 2≤R<4; C4, 4≤R<6; C5, 6≤R<10; C6, R≥10mmh$^{-1}$. Such classification is
based on two considerations: firstly, the number of observation samples in different
rainfall rates roughly conformed to a normal distribution; and secondly, the mean
maximum diameter interval of different rainfall rates gradually increases (Li et al.,
2019). Of course, other studies about classification are referenced and the fact that the
rain rate in this area is smaller than that in the southern China is taken into account (Ma
et al., 2019b; Zeng et al., 2021). Figure 4 shows the mean DSDs at each rain rate class
for six sites. Table 3 contains the number of samples and statistical values of the DSD



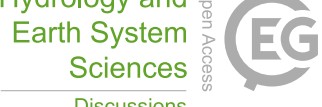

parameters for six classes. Obviously, with the rain rate class increasing, the number
concentration of almost all raindrop sizes and the width of DSD shapes increase, thus
the tail of DSD shape gradually moves towards a larger diameter, which are similar to
the previous studies such as Ma et al. (2019b) and Pu et al. (2020). Taking a number
concentration of 0.01 $m^{-3}mm^{-1}$, the mean maximum diameter of DSD in each class is in
order: 2.3-2.5, 3.2-3.4, 3.9-4.5, 4.3-5.0, 5.0-5.6 and 6.0-7.0 mm (The sixth-class
diameter range is not fully shown in the figure). In class C1, the number concentrations
are relatively similar in different sites; starting from class C2, the differences of number
concentration increase when the diameter is greater than 2mm for 6 sites; and the
differences of number concentration are gradually reflected on each raindrop size bin
as rain rate class increasing. Observingly, the DSDs of BLG, HS and TL have larger
number concentrations in different rain rate class, and the DSD parameters and standard
deviations (SD) are larger, especially for BLG.
Table 3 Statistical of several integral DSD parameters for six rain rate classes at 6 sites.

| Class | Sites | Samples | $Log_{10}N_w$ ($m^{-3}mm^{-1}$) | | $D_m$ (mm) | | $\mu$ | | $Log_{10}N_t$ ($m^{-3}$) | | $R$ ($mmh^{-1}$) | | $Z$ dBZ | |
|---|---|---|---|---|---|---|---|---|---|---|---|---|---|---|
| | | | ME | SD | ME | SD | ME | SD | ME | SD | ME | SD | ME | SD |
| C1(<0.5 mm $h^{-1}$) | LB | 6520 | 3.25 | 0.41 | 0.88 | 0.18 | 12.36 | 7.09 | 1.74 | 0.34 | 0.20 | 0.13 | 12.68 | 4.52 |
| | HS | 10753 | 3.43 | 0.44 | 0.81 | 0.17 | 12.01 | 7.03 | 1.89 | 0.37 | 0.20 | 0.13 | 11.90 | 4.54 |
| | TL | 7858 | 3.52 | 0.44 | 0.79 | 0.16 | 12.91 | 7.12 | 1.96 | 0.37 | 0.20 | 0.13 | 11.78 | 4.16 |
| | SD | 5772 | 3.34 | 0.43 | 0.85 | 0.18 | 11.72 | 6.99 | 1.82 | 0.36 | 0.20 | 0.13 | 12.51 | 4.40 |
| | BLG | 10073 | 3.50 | 0.48 | 0.79 | 0.17 | 12.94 | 7.28 | 1.94 | 0.40 | 0.20 | 0.13 | 11.73 | 4.26 |
| | DLD | 6891 | 3.51 | 0.43 | 0.79 | 0.15 | 13.04 | 6.92 | 1.96 | 0.36 | 0.21 | 0.13 | 12.14 | 4.15 |
| C2(0.5~2 mm $h^{-1}$) | LB | 3318 | 3.66 | 0.41 | 1.06 | 0.24 | 9.93 | 5.75 | 2.30 | 0.28 | 1.00 | 0.41 | 22.55 | 3.27 |
| | HS | 5700 | 3.82 | 0.39 | 0.97 | 0.21 | 10.21 | 5.88 | 2.44 | 0.26 | 0.96 | 0.37 | 21.67 | 3.09 |
| | TL | 5368 | 3.87 | 0.42 | 0.98 | 0.23 | 10.35 | 6.15 | 2.49 | 0.26 | 1.07 | 0.41 | 22.18 | 3.33 |
| | SD | 3778 | 3.73 | 0.41 | 1.03 | 0.23 | 9.94 | 6.14 | 2.36 | 0.28 | 1.02 | 0.40 | 22.40 | 3.15 |
| | BLG | 6411 | 3.97 | 0.47 | 0.94 | 0.25 | 11.24 | 6.72 | 2.56 | 0.30 | 1.07 | 0.43 | 21.69 | 3.69 |
| | DLD | 4778 | 3.88 | 0.37 | 0.95 | 0.20 | 10.91 | 6.02 | 2.47 | 0.24 | 1.01 | 0.40 | 21.60 | 3.19 |
| C3(2~4 mm $h^{-1}$) | LB | 782 | 3.71 | 0.47 | 1.31 | 0.37 | 7.33 | 4.28 | 2.52 | 0.29 | 2.77 | 0.56 | 29.54 | 2.87 |
| | HS | 884 | 3.96 | 0.50 | 1.16 | 0.34 | 8.42 | 5.22 | 2.73 | 0.27 | 2.76 | 0.54 | 28.33 | 3.06 |
| | TL | 1232 | 4.00 | 0.47 | 1.13 | 0.33 | 8.70 | 5.93 | 2.75 | 0.23 | 2.68 | 0.53 | 28.07 | 3.16 |
| | SD | 812 | 3.89 | 0.44 | 1.19 | 0.27 | 8.57 | 5.53 | 2.63 | 0.26 | 2.71 | 0.53 | 28.41 | 2.68 |
| | BLG | 1865 | 4.05 | 0.49 | 1.11 | 0.30 | 8.62 | 5.75 | 2.81 | 0.25 | 2.70 | 0.53 | 27.99 | 3.29 |
| | DLD | 1111 | 3.91 | 0.44 | 1.18 | 0.29 | 7.81 | 5.45 | 2.70 | 0.23 | 2.74 | 0.54 | 28.73 | 3.09 |
| C4(4~6 mm $h^{-1}$) | LB | 229 | 3.80 | 0.47 | 1.41 | 0.40 | 7.33 | 3.94 | 2.65 | 0.31 | 4.76 | 0.57 | 32.69 | 2.63 |
| | HS | 191 | 4.03 | 0.54 | 1.28 | 0.47 | 7.54 | 4.42 | 2.86 | 0.27 | 4.80 | 0.56 | 31.70 | 3.34 |
| | TL | 213 | 3.84 | 0.56 | 1.41 | 0.51 | 6.23 | 4.64 | 2.77 | 0.28 | 4.77 | 0.54 | 32.82 | 3.54 |
| | SD | 187 | 4.03 | 0.41 | 1.24 | 0.27 | 8.35 | 5.02 | 2.80 | 0.22 | 4.76 | 0.54 | 31.32 | 2.52 |
| | BLG | 321 | 3.99 | 0.66 | 1.33 | 0.53 | 7.97 | 6.10 | 2.93 | 0.27 | 4.78 | 0.54 | 32.44 | 4.40 |
| | DLD | 270 | 3.92 | 0.53 | 1.35 | 0.47 | 6.50 | 4.80 | 2.83 | 0.25 | 4.83 | 0.56 | 32.55 | 3.47 |
| C5(6~10 mm $h^{-1}$) | LB | 167 | 3.81 | 0.46 | 1.55 | 0.44 | 6.46 | 3.38 | 2.72 | 0.27 | 7.66 | 1.22 | 35.74 | 2.85 |
| | HS | 49 | 3.69 | 0.74 | 1.70 | 0.68 | 6.89 | 4.82 | 2.75 | 0.38 | 7.42 | 1.09 | 36.14 | 4.29 |





| | | | | | | | | | | | | | | |
|---|---|---|---|---|---|---|---|---|---|---|---|---|---|---|
| | TL | 103 | 3.57 | 0.62 | 1.78 | 0.66 | 5.20 | 4.62 | 2.71 | 0.32 | 7.32 | 1.02 | 37.03 | 3.76 |
| | SD | 128 | 3.96 | 0.39 | 1.42 | 0.35 | 7.10 | 3.96 | 2.82 | 0.21 | 7.68 | 1.17 | 34.76 | 2.42 |
| | BLG | 138 | 3.97 | 0.76 | 1.51 | 0.80 | 8.34 | 6.35 | 2.99 | 0.27 | 7.37 | 1.02 | 35.09 | 4.96 |
| | DLD | 122 | 3.90 | 0.46 | 1.46 | 0.34 | 6.13 | 4.20 | 2.86 | 0.26 | 7.29 | 1.11 | 35.32 | 2.88 |
| $C6(>10 \text{ mm h}^{-1})$ | LB | 87 | 3.85 | 0.44 | 1.73 | 0.53 | 5.08 | 3.05 | 2.87 | 0.32 | 14.81 | 7.57 | 39.58 | 3.57 |
| | HS | 42 | 3.60 | 0.65 | 2.19 | 0.92 | 6.74 | 5.27 | 3.00 | 0.28 | 21.69 | 9.91 | 42.93 | 6.11 |
| | TL | 40 | 3.16 | 0.69 | 2.69 | 1.19 | 4.34 | 5.20 | 2.74 | 0.32 | 18.25 | 9.69 | 44.70 | 5.41 |
| | SD | 59 | 3.66 | 0.29 | 2.04 | 0.46 | 3.30 | 2.48 | 2.91 | 0.16 | 21.07 | 8.34 | 42.85 | 4.10 |
| | BLG | 53 | 3.38 | 0.93 | 2.58 | 1.52 | 5.58 | 6.19 | 3.00 | 0.37 | 21.95 | 9.05 | 44.08 | 7.50 |
| | DLD | 58 | 3.82 | 0.47 | 1.80 | 0.46 | 6.64 | 4.12 | 2.84 | 0.28 | 16.58 | 7.21 | 40.13 | 3.53 |

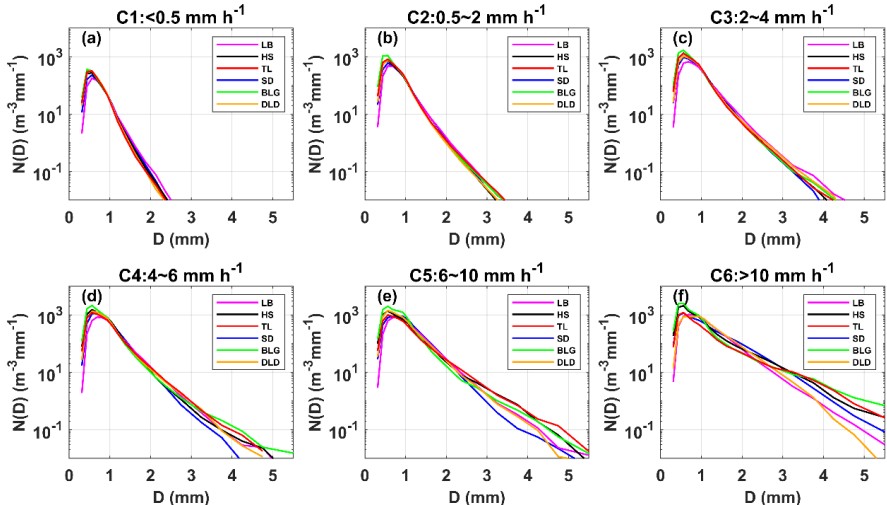


Fig.4 Distribution of mean measured DSD for different rain rate classes at 6 sites.
Fig. 5 shows box-whisker plots of the normalized intercept parameter $\log_{10}N_w$ and
mass-weighted mean diameter $D_m$ for 6 sites at each rain rate class. The middle line in
the box indicates the median. The left and right lines in the box indicate the 25th and
75th. The left and right ends of whiskers indicate the most extreme data points between
5th and 95th, except outliers. The median of $D_m$ gradually increases with a larger value
range when the rain rate class increases, particularly for HS and BLG at class C5 and
C6. The median of $\log_{10}N_w$ increases at class C1 to C3 and then tends to decrease at
class C5 to C6, which the reduction is obvious at sites with a larger value range, such
as HS and BLG. Ma et al. (2019b) also obtains similar conclusions about $D_m$ and
$\log_{10}N_w$. It is indicated that the increase of rain rate is mainly due to the growth in
raindrop size. And the change of number concentration may be caused by the imbalance
between the loss of number concentration at small raindrop size and the addition at
large raindrop size, which implies in a sense that the relation of collision-coalescence
and break-up of raindrops. It is worth noting that the microphysical processes are quite
different among the sites, which are greatly influenced by the surrounding environment.
Because HS and BLG are located on the inside mountains and close to ridge, thus their
dynamics and thermodynamics as well underlying surface are different from other



districts.

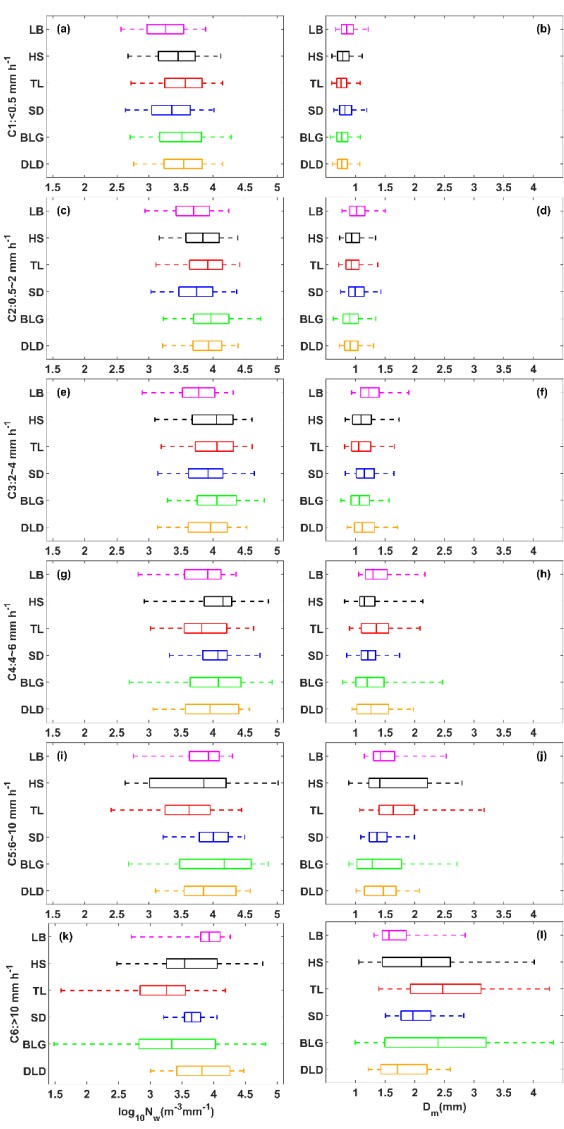


Fig.5 Variation of normalized intercept parameter $\log_{10}N_w$ (a) and the mass-weighted
mean diameter $D_m$ (b) for different rain rate classes at 6 sites. The three lines in box are
25th, 50th and 75th percentiles from left to right, respectively. The whiskers on the left
end and right end are 5th and 95th percentiles, respectively. The colors represent 6 sites
same as other figures.

Figure 6 displays the contribution of different rain rate classes to the total rainfall
at different sites. It is clear that C2 contributes the most to the total rainfall of all sites,
followed by C3, and the sum of two classes of contribution could reach 60% to the total
rainfall. Compared with the districts on the inside and southern slopes, C2 and C3



contribute slightly less to LB and SD sites (i.e. the northern slopes), while C5 and C6
contribute relatively more to LB and SD sites, indicating that there is a greater
probability of heavy precipitation events on the northern slopes. The DSD parameters
in Table 3 provide a more detailed representation of the rainfall differences between the
three geographic locations of the Qilian Mountains, namely the inside, southern slopes
and northern slopes. Meanwhile, it also reflects the characteristics of rainfall on the
eastern and middle sections, such as the eastern section has larger Z and $D_m$ and smaller
$\log_{10}N_w$ and $\log_{10}N_t$ compared to the middle section. It is possible that there is a certain
spatial connection between precipitation at the sites, which is related to the factors like
the source of precipitation vapor, weather system and so on.

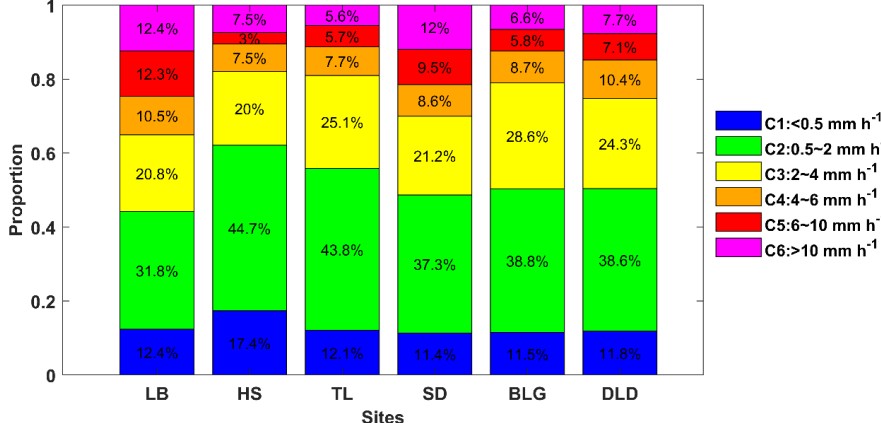


Fig.6 Proportion of rainfall with different rain rate classes to rain amount at 6 sites.

**3.4 DSD properties for different rain types**

Previous studies on DSD have shown that there are significant differences in the
DSD of convective and stratiform rainfall in the same climatic region, which has a great
impact on the parameterization of NWP and remote sensing observations (Bringi et al.,
2003; Penide et al., 2013). Due to the different physical mechanisms of convective and
stratiform rainfall, it can be allowed to discuss the differences of microphysical
structures for rainfall types through their DSD. In some studies, there have been many
classification methods for rainfall types, like Testud et al. (2001) used rain rate; Chen
et al. (2013) combined rain rate and its standard deviation (SD); and Das et al. (2018)
were based on rain rate and radar reflectivity factor. The method from Chen et al. (2013)
was always used to establish samples of convective and stratiform rainfall, in which the
studies' area were concentrated in semi-humid or humid regions with relatively high
rain rate and rainfall. However, Qilian Mountains are located in the semi-arid regions
of China and far from the sea, which the average rainfall rain and rainfall are quite
different from the semi-humid regions. The paper therefore proposes a new
classification method for precipitation applicable to the arid and semi-arid regions of
northwest China based on the classification ideas of Chen and Saurabh.
Firstly, the sequences of DSD with continuous 1-min samples more than 10



minutes are determined, and Rt is defined to denote the rain rate at time t. The first case:
the R of samples from $R_{t-5}$ to $R_{t+5}$ are all less than 5mmh$^{-1}$ and their standard deviation
(SD) is less than 1.5 mmh$^{-1}$; the second case: the R of samples from $R_{t-5}$ to $R_{t+5}$ are
greater than or equal to 5 mmh$^{-1}$ with more than 9 samples and their SD is greater than
1.5 mmh$^{-1}$; the third case: same as the second case but their SD is less 1.5 mmh$^{-1}$.
Secondly, samples satisfying Z<20 and W<0.08 in the second case are removed (Thurai
et al., 2016; Das et al., 2018). And then, samples with $R_t$ great than or equal to 5 mmh$^{-1}$
$^{1}$ in the second case are regarded as convective rainfall and samples with $R_t$ less than 5
mmh$^{-1}$ in the second case are regarded as transition rainfall (the rainfall stage in which
convective precipitation develops and declines). Samples in the first case are regarded
as stratiform rainfall. Through experiments, the third case does not exist.
The $\log_{10}N_w$ and $D_m$ of different rainfall types are different, which make as the
main research objects. Figure 7 shows the variation of $\log_{10}N_w$ with the $D_m$ at different
sites. The blue, red, and yellow scattered points represent stratiform, convective and
transition rainfall, respectively. Obviously, there are fairly clear boundaries between the
scatter points for different precipitation type events and the same dividing line can be
used to distinguish different rainfall types at different sites. The black solid lines were
drawn based on visual examination of the data with a slope of approximately -1.60 and
intercept of 6.008 to represent the split between stratiform, transition and convective
rainfall in all subplots. The black dashed line can distinguish transition rainfall
(transition and stratiform rainfall have overlap area) with a slope of approximately -
3.338 and intercept of 6.847. Note that the dividing line between stratiform and
convective rainfall has the same slope obtained by Bringi et al. (2003) (solid green line
with a slope of -1.6 and intercept of 6.3) who fitted the composite results based on
disdrometer data and from radar retrievals covering many climate conditions from near
equator to plateau. The $\log_{10}N_w$ and $D_m$ from the figures to stratiform, convective and
transition rainfall are respectively concentrated in 3.1-3.9 m$^{-3}$mm$^{-1}$, 0.75-1.1 mm; 3.8-
4.2 m$^{-3}$mm$^{-1}$, 1.4-1.6 mm; 3.6-4.0 m$^{-3}$mm$^{-1}$, 1.05-1.2 mm. Compared to the maritime-
like cluster and continental-like cluster of convective rainfall proposed by Bringi et al.
(2003), the convective events in Qilian Mountains are more consistent with the
continental-like cluster (the gray rectangle with smaller $\log_{10}N_w$ and larger $D_m$ in Figure
7). There are isolated convective events in the maritime-like cluster, but it is difficult to
have more events from the trend between $\log10N_w$ and $D_m$. This is also consistent with
features of geographical location in Qilian Mountains.

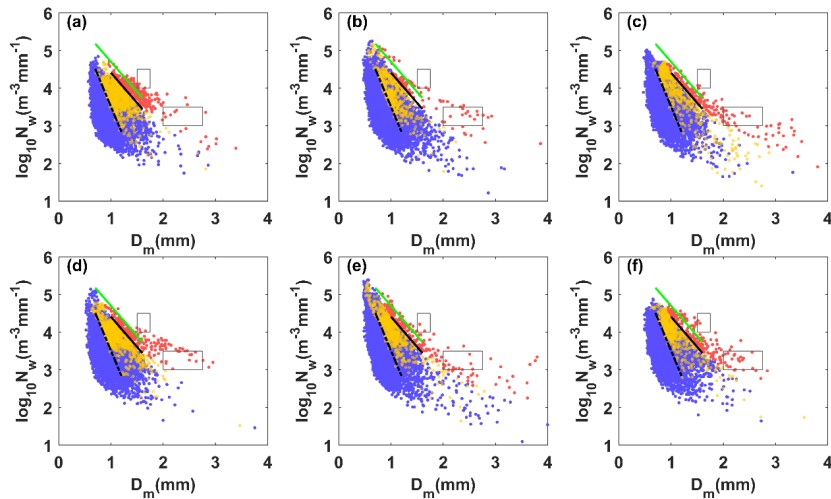


Fig.7 Scatter plot of $\log_{10}N_w$ versus Dm for different rain types at (a) LB, (b) HS, (c)
TL, (d)SD, (e)BLG, (f)DLD. The stratiform cases, convective cases and transition cases
are represented by blue, red and yellow circle dots, respectively. The black dashed lines
are the $\log_{10}N_w$-$D_m$ relationship for stratiform versus convective cases and stratiform
versus transition case.

Figure 8 shows the mean DSDs for stratiform, convective and transition rainfall at
six sites. The range of number concentrations and corresponding raindrop diameters for
the three types are significantly different, matching the basic characteristics of DSD.
The mean DSDs of stratiform rainfall differ slightly among sites; convective rainfall
has big differences at sites; and transition rainfall appears more differences beginning
at larger than 2.2 mm diameter, which are the expected results. Stratiform rainfall
usually has a large horizontal extent and a homogeneous cloud distribution, which
makes the DSD characteristics basically same under the influence of same cloud system
in the mountainous areas. But convective rainfall is related to the local thermal and
dynamical factors, which could lead to differences in the DSD at different sites adding
the complex topography and diverse underlying in mountainous areas. For example, in
convective rainfall, there is a significant increase in the number concentration of
raindrops larger than 2.2 mm diameter at BLG, HS and TL, indicating that these districts
are conducive to the development of convective precipitation. And the number
concentration of small raindrops in BLG and HS is higher than that in TL (the southern
slope), which may be due to the higher altitude of the inside sites reducing the falling
distance of raindrops after exiting the cloud and decreasing the impact of collision on
the raindrop evolution. In other words, even in the same rainfall type, the microphysical
process of rainfall at different sites is still different, depending on the topography and
position of the observation point relative to the cloud base.



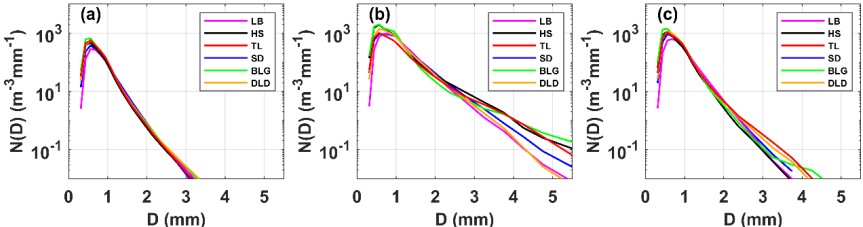


Fig.8 Distribution of mean measured DSD for (a) stratiform rainfall, (b) convective
rainfall and (c) transition rainfall at 6 sites.
Figure 9 is the box-whisker plots of $\log_{10}N_w$ and $D_m$ for different rain types. The
$\log_{10}N_w$ and $D_m$ of stratiform rainfall are smaller than that of convective rainfall but
larger than that of transition rainfall. Sites with a large $\log_{10}N_w$ value range have a larger
values range for $D_m$; and sites with a large median for $\log_{10}N_w$ have a smaller median
for $D_m$, especially at HS and BLG sites in convective rainfall. Based on the mean value
of six sites in Table 4, the DSD characteristic in Qilian Mountains consists of a larger
$N_w$ and a smaller $D_m$ due to melting of tiny, compact graupel, and rimed ice particles
(relative to large, low-density snowflakes). Compared with transition rainfall, the $D_m$
of convective rainfall is obviously larger, indicating that the increase in rain rate in this
area is mainly due to the growth in raindrop size. Moreover, the northern slopes should
consider the increase of number concentration, because the $\log_{10}N_w$ of convective
rainfall also have increased. Note that the number of convective samples on the northern
slope is higher than that of other sites, which correspond to the speculation in the
contribution of different rain rate classes. On average of stratiform rainfall, the
dispersion degree of $\log_{10}N_w$ and $D_m$ in different sites is 8.3% and 10.0%, respectively;
and convective rainfall is 10.4%、23.4%, respectively. The standard deviations of DSD
parameters at HS and BLG sites are relatively large.
Table 4 Statistical of several integral DSD parameters for six sites with stratiform
rainfall, convective rainfall and transition rainfall

| Type | Sites | Sample | $\log_{10}N_w$ (m⁻³mm⁻¹) | | $D_m$ (mm) | | $\mu$ | | $\log_{10}N_t$ (m⁻³) | | R (mmh⁻¹) | | Z dBZ | |
|---|---|---|---|---|---|---|---|---|---|---|---|---|---|---|
| | | | ME | SD | ME | SD | ME | SD | ME | SD | ME | SD | ME | SD |
| S | LB | 7123 | 3.42 | 0.42 | 0.96 | 0.21 | 11.48 | 7.98 | 1.98 | 0.38 | 0.54 | 0.60 | 16.93 | 5.93 |
| | HS | 12694 | 3.60 | 0.44 | 0.88 | 0.21 | 11.24 | 7.89 | 2.14 | 0.40 | 0.54 | 0.58 | 16.17 | 6.06 |
| | TL | 10091 | 3.71 | 0.43 | 0.87 | 0.20 | 11.90 | 8.01 | 2.23 | 0.39 | 0.65 | 0.67 | 16.85 | 6.15 |
| | SD | 7175 | 3.51 | 0.44 | 0.95 | 0.22 | 11.15 | 8.03 | 2.07 | 0.39 | 0.62 | 0.64 | 17.36 | 6.10 |
| | BLG | 12467 | 3.72 | 0.49 | 0.88 | 0.23 | 12.24 | 8.50 | 2.25 | 0.44 | 0.70 | 0.74 | 17.11 | 6.33 |
| | DLD | 9685 | 3.70 | 0.42 | 0.88 | 0.21 | 11.91 | 7.91 | 2.23 | 0.38 | 0.67 | 0.69 | 17.18 | 6.13 |
| C | LB | 292 | 3.91 | 0.35 | 1.49 | 0.35 | 6.50 | 3.30 | 2.81 | 0.23 | 9.28 | 5.56 | 35.88 | 3.59 |
| | HS | 100 | 3.85 | 0.67 | 1.71 | 0.84 | 6.33 | 4.33 | 2.95 | 0.30 | 12.55 | 13.75 | 37.32 | 6.64 |
| | TL | 159 | 3.54 | 0.59 | 1.87 | 0.74 | 5.21 | 4.97 | 2.72 | 0.30 | 9.48 | 6.91 | 37.96 | 5.21 |
| | SD | 219 | 3.91 | 0.37 | 1.54 | 0.47 | 6.61 | 4.68 | 2.85 | 0.19 | 10.75 | 7.68 | 36.24 | 5.02 |





|  |  |  |  |  |  |  |  |  |  |  |  |  |  |
|---|---|---|---|---|---|---|---|---|---|---|---|---|---|
|  | BLG | 198 | 3.91 | 0.74 | 1.64 | 0.97 | 8.00 | 7.37 | 3.00 | 0.27 | 10.57 | 15.49 | 36.29 | 6.75 |
|  | DLD | 203 | 3.94 | 0.48 | 1.50 | 0.43 | 6.96 | 5.24 | 2.87 | 0.27 | 9.41 | 6.04 | 35.89 | 4.27 |
| T | LB | 787 | 3.76 | 0.39 | 1.15 | 0.21 | 8.37 | 4.35 | 2.47 | 0.31 | 2.16 | 1.25 | 26.42 | 3.89 |
|  | HS | 541 | 3.89 | 0.49 | 1.05 | 0.29 | 8.98 | 6.74 | 2.59 | 0.33 | 1.81 | 1.15 | 24.79 | 3.89 |
|  | TL | 465 | 3.77 | 0.70 | 1.22 | 0.49 | 8.81 | 6.91 | 2.56 | 0.44 | 2.30 | 1.21 | 27.10 | 4.39 |
|  | SD | 819 | 3.87 | 0.41 | 1.12 | 0.26 | 8.23 | 5.46 | 2.59 | 0.28 | 2.28 | 1.18 | 26.59 | 4.04 |
|  | BLG | 665 | 4.04 | 0.51 | 1.04 | 0.31 | 10.33 | 7.31 | 2.72 | 0.33 | 2.19 | 1.13 | 25.66 | 4.44 |
|  | DLD | 503 | 3.95 | 0.46 | 1.10 | 0.30 | 8.69 | 6.16 | 2.67 | 0.31 | 2.35 | 1.17 | 26.60 | 4.20 |

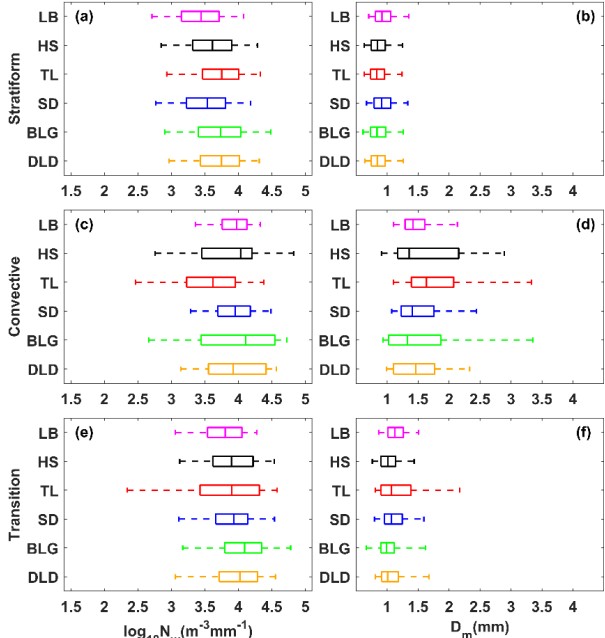


Fig.9 Same as Fig. 5 but for different rain types at 6 sites.
**3.5 Implications for radar rainfall estimation with DSD**
The sixth moment of raindrop diameter is proportional to the radar reflectivity
factor and the 3.76 moment is approximately rain rate (they can be calculated by
Equations 4 and 5). Generally, the theoretical basis of the QPE for single polarization
radar (ground based or space based) is the power relationship between radar reflectivity
and rainfall rate ($Z=AR^b$). This makes the coefficients A and exponents b of the power
relationship heavily dependent on the variation of the DSD. Therefore, it is necessary
to obtain the A and b of different sites according to different rainfall types.
Figure 10 shows the Z-R scatter plots for different sites and the fitted power-law
relationships for different rainfall types. The blue and red scatters represent stratiform
and convective rainfall, respectively. The purple, red and black solid lines indicate Z-
R relationships for stratiform, convective and total rainfall, respectively. It shows that Z-
R scatters for HS and BLG are relatively scattered around 5mmh-1 rain rate. Besides,
the Z-R relationship of total rainfall underestimates stratiform rainfall at low R values





and underestimates convective rainfall at high R values. On the average of Z-R
relationship using a least-squares method, the dispersion degree of A and b in different
sites is 42.5% and 10.7%, respectively, which reveal the large differences in mountains.

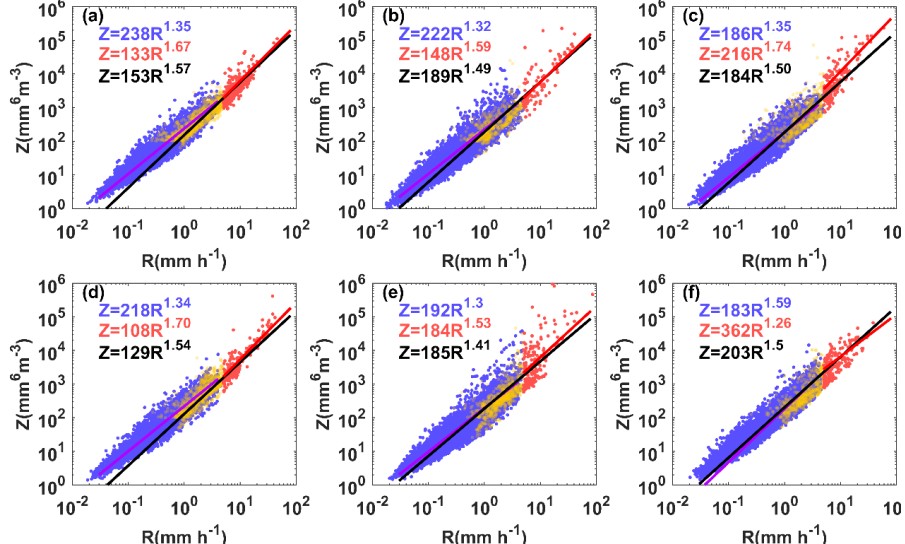


Fig.10 Scatter plot of Z (mm$^6$m$^{-3}$) versus R (mmh-1) for three rain types at (a) LB, (b)
HS, (c) TL, (d)SD, (e)BLG, (f)DLD. The blue, red and yellow circle dots, respectively,
stand for stratiform, convective and transition cases. The purple, red and black lines
denote the Z-R relation. The blue, red and black formula denote stratiform, convective
and total Z-R relationships.

In order to compare the six sites Z-R relationship with some standard Z-R
relationships, Z=300R$^{1.4}$ for convective rainfall commonly used on radar and Z=200R$^{1.6}$
(i.e. M48) for stratiform rainfall commonly used on midlatitude areas are provided in
figure 11. Overall, convective rainfall has smaller values of A and larger values of b
than that of stratiform rainfall (excluding DLD). The A values of convective rainfall are
smaller than the commonly used Z-R relationship with large differences, but the b
values are greater. The distribution of A and b for stratiform rainfall is relatively
concentrated with A and b ranging from 186-238 and 1.3-1.35, respectively. The A
values of SR are close to the M48, and the b values are close to and smaller than the Z-
R of global SR. The DLD station has a similar Z-R in stratiform rainfall with M48,
while its convective rainfall is different from other sites with a larger A value (twice as
large as other sites) and smaller b value. In addition, it can make it clear that the A value
of stratiform rainfall increases from the southern slopes to northern slopes, while the
convective rainfall is opposite. And the Z-R relationships of the same side are more
consistent, such as both on inside or the northern slopes, which have geographic
characteristics.

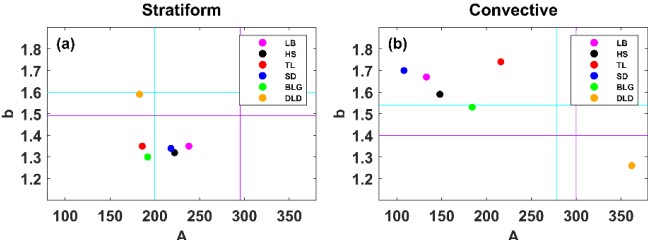


Fig.11 A and b values of the Z-R relationship for (a) stratiform rainfall and (b)
convective rainfall at 6 sites. The purple lines in Fig. 12a and 12b correspond to the
global Z-R model ($Z = 295R^{1.49}$ for continental stratiform rainfall and $Z = 278R^{1.54}$ for
convective rainfall, respectively) (Ghada et al., 2018). The cyan line in Fig. 12a
represents midlatitude stratiform rainfall Z-R model ($Z = 200R^{1.60}$, Marshall, 1948); the
cyan line in Fig. 12b represents the convective rainfall Z-R model ($Z = 300R^{1.40}$) applied
to the operational weather radar (Fulton et al., 1998).
**4  Discussion**

The paper analyses the statistical characteristics of DSD at different sites in the
Qilian Mountains during the rainy season, which not only contain rainfall classes and
rainfall types but more importantly reflect the differences between different sites. The
results from different aspects can be mutually confirmed and have a good representation
of the spatial distribution, making as a great factual basis for the discussion of the
microphysical structure for precipitation. For example, with the rain rate class rising,
the number concentration of all size bins is increased and the width of DSDs become
wider, which as a feature are manifested in rain types that convective rainfall has a
larger rain rate. In terms of spatiality, the characteristics of precipitation on the inside
and southern slope are closer, whether the overall DSD or the DSD parameter
distribution. But there are some obvious variabilities in the inside mountains for DSD
parameters due to the influences of its local dynamics and thermal. On the other hand,
these characteristics also exhibit some differences between the middle and eastern
sections in Qilian Mountains, especially in the discussion of DSD parameters for
rainfall classes and rainfall types (shown as Figures 5 and 9). This spatial variation in
DSD suggests that microphysical processes in DSD are influenced by complex
topography (altitude, mountain alignment) and potentially related to the source of water
vapor, development of precipitation process and anthropogenic factors.

Compared to the precious studies that are focused on eastern, southern and
northern China as well Tibetan Plateau, the Qilian Mountains have its own unique DSD
characteristics and Z-R relationship during the rainy season, which include the smaller
raindrop diameter with higher number concentration. Moreover, the division of rain rate
classes in Qilian Mountains more adequately reflects the DSD characteristics at each
class, unlike using the classification method of other sites with larger rain rates. Above
all, it is Qilian Mountains that the proposed classification of stratiform and convective
rainfall is applicable to, which is located on the arid and semi-arid regions.

As aforementioned, the characteristics of DSD mainly describe on the diameters


larger than 0.2 mm, which are limited by the observation instruments that cannot detect
the small drops on diameter less than 0.2 mm. So, it is not a complete DSD and
underestimates the number concentration of small drops on diameter less than 0.5 mm.
Recent studies have been devoted to improving DSD observations in order to overcome
the limitations of disdrometer. A study by Thurai et al. (2017) have obtained a more
complete DSD by splicing the 2DVD and MPS (Meteorological Particle Spectrometer)
to observe DSD and developed a technology to reconstruct the drizzle mode DSD
(Raupach et al., 2019), which has a good presentation to the DSD of small raindrops
and more important applications.

## 5   Summary and conclusion

Based on the six-months DSD data observed in the southern slopes, northern
slopes and inside of Qilian Mountains, the characteristics and their differences of DSD
are studied, and Z-R relationships of six districts are discussed. The main conclusions
are as follows.
For small raindrops, the number concentrations on the inside and southern slopes
districts are greater than that on the northern slopes; for midsize raindrops, the number
concentrations decrease sequentially on the northern slopes, southern slopes and inside
districts; for large raindrops, the number concentrations on the inside districts are larger.
In addition, the number concentrations of raindrops in the middle section of the
mountainous area is slightly greater than that in the eastern section.
1.  For all rainfall events, the number concentration of small and large raindrops on
the inside and southern slopes are greater than that on the northern slope, while
midsize raindrops are less. The DSD of inside mountains has a great variability,
which is quite different from the northern slope.
2.  The DSDs are divided into six categories based on rainfall rate: C1, R<0.5; C2,
0.5≤R<2; C3, 2≤R<4; C4, 4≤R<6; C5, 6≤R<10; C6, >10 mm h-1. As the rain
rate increases, the median of $D_m$ for each station is gradually larger and the median
of $N_w$ rises on C1-C3 and then decreases on C4-C6, as well the differences of
number concentration on each drop size increases. Especially in the inside
mountains. The most contribution to the total rainfall at different sites is C2 class
and C3 class next, with the sum of contribution reaching 60%. Besides, the C5 and
C6 class have a relatively large contribution to the north slope with a greater
probability of heavy precipitation events.
3.  There is a rather clear boundary in the distribution of $log_{10}N_w$ versus $D_m$ between
the rainfall types, which the split line between stratiform and convective rainfall
has the same slope with the line given by Bringi et al. The dispersion degree of
$log_{10}N_w$ and $D_m$ at sites are 8.3% and 10.0% for stratiform rainfall and 10.4% and
23.4% for convective rainfall, respectively. The standard deviations of DSD
parameters on inside sites are larger, making it easier to increase the number
concentration of large raindrops in convective rainfall.
4.  The Z-R relationships of different sites in stratiform rainfall are similar and
generally underestimated by the $Z=200R^{1.6}$ model used to the midlatitude
stratiform rainfall; the Z-R relationships for convective precipitation vary greatly





at different station, which are overestimated by $Z=300R^{1.4}$ at lower rain rates
values and underestimated at higher rain rates values. The dispersion degree of
coefficient A and exponent b in Z-R relationship for sites are 42.5% and 10.7%,
respectively. Overall, the A value of stratiform rainfall increases from the southern
slopes to northern slopes, while the convective rainfall is opposite. And the Z-R
relationships of the ipsilateral sites are more consistent.
5.    The analysis of DSD and DSD parameters can reflect the characteristics of the
southern slope, northern slope and inside sites, as well as the differences between
the eastern and middle sections of Qilian Mountains.
This study reveals the microphysical variability of precipitation in the complex
topography of the arid and semi-arid regions of Northwest China, which can not only
improve local numerical simulations, but also provides a basis for further understanding
of the differences in DSD characteristics formed at mesoscale due to topographic
factors and water vapor distribution, etc. It is important to note that this should be one
of the fundamental studies for the future implementation of weather modification,
which is of great significance to solving the shortage of water resources in the arid and
semi-arid regions.
*Data availability.* Disdrometer data used in this study are available by contacting the
authors.
*Author contributions.* WM conducted the detailed analysis; WZ provided financial
support and conceived the idea; MK collated the observation data; all the authors
contributed to the writing and revisions.
*Competing interests.* The authors declare that they have no conflict of interest.
**Acknowledgments**
The work was supported by Weather modification ability construction project of
Northwest China under grant No. ZQC-R18208 and The Second Tibetan Plateau
Comprehensive Scientific Expedition Grant No. 2019QZKK0104.



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
