# Peer review of "Statistical characteristics of raindrop size distribution during rainy seasons in Complicated Mountain Terrain"

_Hydrology and Earth System Sciences, 2022_

## Author Response (AR1)

**Response to Reviewers**

Dear Editors and Reviewers:

Thanks for giving us an opportunity to revise our manuscript and the reviewers' comments concerning our manuscript. Those comments are valuable and very helpful for revising and improving our paper, as well as the important guiding significance to our researches.

After receiving the comments, we attached great importance to them and carefully discussed the issues mentioned in the manuscript. Though this period of thinking, we thoroughly revised the manuscript and improved every point, which we hoped meet with approval. Revised portion are marked in the paper. The main corrections in the paper and the responds to the reviewer's comments are as flowing.

We highly appreciate your time and consideration to allow us resubmit a revised copy of the manuscript. Please let us know if there is anything need to discuss during the review process.

Authors,

Sincerely.

**Response:**

Reviewer 1

Raindrop size distribution and the number of raindrops is an important parameter to describe the microstructure of precipitation. Numerous studies have been carried out the statistical characteristics of DSD in different regions. Qilian mountains are the vitally important ecological protection barrier and important water source in northwest arid areas of China. In this paper, the authors select 6 sites with different backgrounds representing the southern slopes, northern slopes and inside of Qilian mountains. This study reveals the microphysical variability of precipitation in the complex topography of the arid and semi-arid regions of Northwest China, which is of great significance to solving the shortage of water resources in the arid and semi-arid regions. The manuscript is of high quality and innovative. The data are full and reliable. I suggest that it be accepted after minor revisions.

Issue 1: The English and grammar of the article need to be carefully revised.

Revision: Thank for the advice. The language of the manuscript has been revised as well as formatting and punctuation.

Issue 2: How to determine the observation instruments are at the same accuracy standard in the 6 sites?

Revision: Thank for the comments. The instruments are used the same type, including the same particle size classification and velocity classification as well as the sensor of observation instruments. Besides, it is also used the same data processing and quality control, which insure the same accuracy at time and particle size.

Issue 3: In Fig 1, the size of sites is small and unclear. Add the photos of observation station or equipment.

Revision: Thank you for reminding us the description. As suggested we have redrawn the diagram (Fig 1) to express the geographical overview of the Qian Mountains and the sites, with clearer google satellite map and more rational selection of drawing areas. Also, we added the photo of equipment placed at one of the observation sites.

Issue 4: The research needs to further highlight the reasons for the differences between sites in the discussion and conclusion. And how is the precipitation different from other areas?

Revision: Thank you for your comments. According to the characteristics of raindrop size distribution (DSD) in Qilian Mountains, we find there are some similarities in different sites, while different from other areas. This is mainly due to melting of tiny, compact graupel, and rimed ice particles (relative to large, low-density snowflakes). Besides, there are also some similarities such as the basic law of stratiform and convective rainfall reflecting in the raindrop size distribution. However, it still exists some differences in Qilian Mountains, especially the DSD parameters, because they have different altitudes and geographical environments. Based on the suggestion and above description, we have supplemented the discussion section on . In order to better illustrate the precipitation difference between Qilian Mountains and other areas, we will choose representative site in Qilian Mountains to compare with other areas.

Issue 5: Extended discussion: Whether the change of DSD is related to other meteorological factors, such as local wind speed?

Revision: Thank for the tips. DSD can reflect the microstructure of precipitation. But it involves a series of microphysical and physical processes from rain generation to falling. There will be more research to explore the possible factors about the change of DSD. And we will continue to think about the contribution of local wind speed on the change.

The authors investigate the characteristics of the raindrop size distribution (DSD) over the complex mountainous terrain Qilian Mountains which are sensitive to climate change in recent decades. Such a study is very helpful to increase the knowledge of the precipitation regimes over the arid and semi-arid region. Overall, the study is written well in terms of science and techniques, and can be accepted and published after minor revision. More comments are as follows:

Issue 1: On line 18, the "which" had better to be replaced with "while".

Revision: Thank for your advice. We have revised the conjunction

Issue 2: The period on line 32 should be updated with English style.

Revision: Thanks for your advice. We have revised the parentheses with English style.

Issue 3: The period before (SR) should be removed on line 50.

Revision: Thanks for your advice. We have deleted the period.

Issue 4: "in southeast" should be updated as "in the southeast".

Revision: Thanks for your advice. We have updated this statement in the whole manuscript.

Issue 5: "results from" had better be replaced with "measurement in" on line 56.

Revision: Thanks for your advice. We have replaced the expression.

Issue 6: Insert a blank space between the number and unit on line 57.

Revision: Thanks for your advice. We have inserted a blank between the number and unit in the whole manuscript.

Issue 7: "vary from location" had better be replaced with "vary with", or "vary from location to location".

Revision: Thanks for your advice. We have revised as "vary from location to location"

Issue 8: The equation on line 139 shows up suddenly and suffers from discontinuity in the context. Similar case can be seen for Eq. (7) on line 169.

Revision: Thanks for your advice. We have added some contexts (line 161 and 205) before the both equations to improve the continuity.

Issue 9: "with" or "by" should be added after the word "calculated" on line 149.

Revision: Thanks for your advice. We have added the word "by" after the word "calculated" and checked the similar problems.

Issue 10: Refine the sentence on 167-168.

Revision: Thanks for your advice. We have refined as "And it has better fitting capability than M-P distribution on the broader variation of DSD fluctuations, including the middle rain drops, especially on small and large rain scale".

Issue 11: Replace "to be well fitted" with "to well fit" on line 167.

Revision: It shown as the above response.

Issue 12: Refine the sentence on line 174-175.

Revision: Thanks for your advice. We have refined as "Although, the gamma distribution is commonly accepted, the normalized gamma distribution has also been widely adopted with its independent parameters and clear physical meaning as follows".

Issue 13: Add legends for different color points, and add descriptions for the rectangles in grey line in the subfigures in Fig.7.

Revision: Thanks for your advice. We have updated the legends and descriptions in Fig 7.

Issue 14: "with the rain rate class rising" can be refined as "as the rainfall rate increases".

Revision: Thanks for your advice. We have refined it and checked in the whole manuscript.

Issue 15: Refine sentence on line 478-479.

Revision: Thanks for your advice. We have refined as "Above all, the proposed classification of stratiform and convective rainfall is suitable for Qilian Mountains, which is applicable to the precipitation in the arid and semi-arid regions.". And we also rewrote this part.

Issue 16: "Fig 1" needs to be considered for better presenting sites information.

Revision: Thanks for your advice. We have changed Fig 1 with bigger size of sites and smaller areas in the map, which better presents the sites information.

Issue 17: The differences from different sites can be described more clearly in the Conclusion section.

Revision: Thanks for your advice. We have revised Conclusion section, which described the differences from four aspects including different rainfall rates and types, as well Z-R relationship.

Issue 18: Some key raindrop parameters can be reported in the Analysis section, such as 3.4 Section reflecting the differences in different rain types.

Revision: Thanks for your advice. We have considered some key raindrop parameters to analyze and compare such as $\log_{10}N_w$ and $D_m$. But there are six sites showing the values, which makes it hard to choose site or the average values of sites. We will add the key parameters of typical site to report the differences in different rain types. And we also prepare another manuscript chosen one site to indicate the differences with other areas. We think it will be more clearly reported with some key raindrop parameters' values.

Issue 19: Line 320: "based on the classification ideas of Chen and Saurabh", Saurabh is not shown in the part of classification method. Please check this sentence.

Revision: Thanks for your comment. "based on the classification ideas of Chen and Saurabh" should be revised as" "based on the classification ideas of Chen et al. (2013) and Das et al. (2018)"

Issue 20: Check the accuracy of the subscripts in the manuscript.

Revision: Thanks for your comment. We have revised the subscripts in the whole manuscript.

**Reviewer 3**

**General comment**

The English is not up to the standard of a journal like EGU-HESS. The manuscript absolutely needs to be checked and improved on this aspect as for some instances, the reader would be confused and can only guess what the authors wish to say. In addition to this, the manuscript has a high frequency of occurrence of typo on units, punctuation and itemization. It thus needs a careful proofreading either by the authors or an external reviewer. I find it hard to concentrate on the content of the paper, and thus would suggest that the authors improve that aspect first, and then

submit a revised version that could be reviewed for assessing the content. I thus suggest a major revision based on this comment only.

Thanks for your comment and the opportunity. We have revised the language of whole manuscript as well as some long or confused sentences, which will be easier to understand. Besides, we checked all the subscripts, units and format in the manuscript and revised them. So, we sincerely hope that the manuscript can move forward in the journal.

The study is based on the HSC-OTT Parsivel2: the authors refer to the OTT and HSC manufacturers of the instrument. It is unclear if this is the exact same instrument as the OTT Parsivel2 found extensively in the DSD literature, or if it is a slightly different version. It would be good if the author can provide more information on this.

Thanks for your advice. The DSG4 disdrometer is produced and sold by Huatron (China), including the sensor is mainly created from OTT Messtechnik (Germany). Essentially, there is not much difference between them (the core components are made by OTT Messtechnik). And after data quality control, the available data have accounted for a high percentage of total number of samples

It would be important to make the DSD data available on a repository. This is predominantly the norm now in the new DSD studies and would help advance science. This is not mandatory as part of the HESS policy (I suppose), but it should be encouraged nevertheless.

Thanks for your affirmation and recognition. We will try our best to improve the manuscript.

**Specific comments**

Issue 1: Line 126 units upper script

Revision: Thanks for your comment. We have revised as "m s-1" and checked the whole manuscript.

Issue 2: Line 131 starting with (1) is inappropriate here.

Revision: Thanks for your comment. We replaced the period before (1) with a colon, and then continued (2), (3), (4), (5) with a semicolon.

Issue 3: Line 131 to 142 you could cite Jaffrain et al. (2011) and Guyot et al. (2019) here:

Jaffrain, J. and Berne, A.: Experimental quantification of the sampling uncertainty associated with measurements from PARSIVEL disdrometers, J. Hydrometeorol., 12, 352–370, https://doi.org/10.1175/2010JHM1244.1, 2011.

Guyot, A., Pudashine, J., Protat, A., Uijlenhoet, R., Pauwels, V. R. N., Seed, A., and Walker, J. P.: Effect of disdrometer type on rain drop size distribution characterisation: a new dataset for south-eastern Australia, Hydrol. Earth Syst. Sci., 23, 4737–4761, https://doi.org/10.5194/hess-23-4737-2019, 2019.

Revision: Thanks for your advice. We have read the two articles and cited them in this part.

Issue 4: Line 138 do no use "can't" in abbreviated form

Revision: Thanks for your advice. We have revised the expression form

Issue 5: Line 157 following equations

Revision: Thanks for your advice. We have revised as "equations".

Issue 6: Line 174 add references on the parameterization of the DSD

Revision: Thanks for your advice. We have added the article from Zhang et al. (2019).

Issue 7: Figure 1 is not up to the standards in terms of resolution

Revision: Thanks for your advice. We have revised Figure 1 and chosen bigger the size of sites, including the clearer google satellite map and more rational selection of drawing areas

Issue 8: Section 3.1 could you provide a summary of the data of each site in terms of the number of samples before and after quality control, and DSD stats (see for example in Angulo-Martinez et al. 2015 or Guyot et al. 2019).

Revision: Thanks for your advice. We have added the number of samples before and after quality control in different sites.

Angulo-MartiÌnez, M., and A. Barros, 2015: Measurement uncertainty in rainfall kinetic energy and intensity relationships for soil erosion studies: An evaluation using PARSIVEL disdrometers in the Southern Appalachian Mountains. Geomorphology, 228, 28-40.

Issue 9: Line 127 spacing between value and units

Revision: Thanks for your comment. We have added the space and checked the whole manuscript.

Issue 10: Figure 3 space between Z and (dBZ); missing "." at the end of the figure caption

Revision: Thanks for your advice. We have added the space and corrected the wrong figure caption in the words.

Issue 11: Figure 11: it would be good to add results from the literature as well on this graph so we can compare the value of the coefficients found in that paper with data from elsewhere (mountainous region, DSD from China in particular).

Revision: Thanks for your advice. In this article, we compared with some common relationships which are widely used in numerical model. There are six sites in this study and it is hard to choose more appropriate site to compare with elsewhere. And other researches also use one site to analyze the characteristics of local area. So we are preparing another manuscript chosen one typical site in Qilian Mountains to indicate the differences with other areas. It would be more clearly reported with some key raindrop parameters' values, as well the Z-R relationship.

Issue 12: Line 516 Bringi et al.: which year?

Revision: Thanks for your comments. We have added the year. It is Bringi et al. (2003).

---

## Author Response (AR3)

**Response to Reviewers**

Dear Editors and Reviewers:

Thanks for giving us a chance to improve our manuscript. Those comments concerning our manuscript are very helpful for revising, as well as the important guiding significance to our researches.

After receiving the comments, we attached great importance to them and carefully discussed the issues mentioned in the manuscript. Though this period of thinking, we thoroughly revised the manuscript and improved these points, which we hoped meet with approval. Meanwhile, it gave us a special opportunity to interact with other scholars who also study on the microstructure of precipitation. Revised portion are marked in the paper. The main corrections in the paper and the responds to the reviewer's comments are as flowing.

We highly appreciate your time and consideration to allow us resubmit a revised copy of the manuscript. Please let us know if there is anything need to discuss during the review process.

Authors,

Sincerely.

**Response:**

Reviewer 1

Thanks for the recognition of the manuscript. We will continue to work on the series of researches. And this opportunity makes a lot of sense for us. If the article can be published, it will be the first one after I graduation. Finally, I would like to thank again for the valuable comments, which have made the article better.

Reviewer 2

Rain drop size distribution over the southern slopes, northern slopes and interior of the Qilian Mountains were analyzed in this article. This article helps us understand the micro physical characteristics of precipitation over the complex mountainous terrain in the arid and semi-arid regions, which is meaningful. However, there are the following problems to be solved at present, so I suggest a major revision of the article before considering it for publication.

*Main comments:*

Issue 1: The influence of different altitudes on DSD is mentioned in line 80, but there is no study or conclusion about it. Please add these contents.

Revision: Thank for the advice. The influence of different altitudes on DSD is understood from the two aforementioned articles. Combination with the geography of the Qilian Mountains, there are certain altitude differences between observation sites (see Table 1), which inspire us to explore this topic. In this manuscript, the DSD differences shown as the results of the southern slopes (average altitude: 2933 m), northern slopes (1845 m) and interior (2398 m) of the Qilian Mountains, firstly prove the existence even at small spatial scale and partly demonstrate the influence of altitude differences. In fact, the sites of interior of the Qilian Mountains are close to ridge from the topography (longitudinal section of mountain). However, because of the limited observation conditions, we would like to have more observation sites to consider this issue.

Issue 2: It is mentioned in line 94 that the main purpose of this study is to improve the accuracy of QPE. However, there is no research to improve QPE in this manuscript. Only the differences of parameters in Z-R relationship are analyzed. What are the implications of the differences for improving QPE accuracy and what specific reference suggestions can it bring to improve QPE accuracy? If possible, please verify the improvement of QPE accuracy through tests, and find out how much improvement?

Revision: Thank for the comments. This part of research is designed mainly from a large number of other studies similar to the DSD. In these studies, the localized Z-R relationships ($Z=AR^b$) calculated from DSD are shown in different research areas, which would replace the general ones like $Z=300R^{1.4}$ or $Z=200R^{1.6}$ (respectively, convective rainfall commonly used in radar and stratiform rainfall commonly used in midlatitude areas). For example, Ma et al. (2019b) obtained the relationship for convective rainfall ($Z=158R^{1.68}$) and stratiform rainfall ($Z=171R^{2.15}$) in Beijing; Zhang et al. (2019) fitted the relationship for convective rain in monsoon season in southern China, with a higher value of A and lower value of b which compared with the standard Z–R relationship; Wang et al. (2021) derived the relationship for convective rainfall ($Z=53.69R^{1.71}$) and stratiform rainfall ($Z=114.79R^{1.34}$) on the Southeast Tibetan Plateau. In a way, choosing the appropriate A and b based on different rain types has a great significance in improving the regional radar QPE. Based on the advice and after careful consideration, we have revised as "refine the local QPE", which hopes it could play a role in radar applications on the study of precipitation estimation for the Qilian mountains.

Issue 3: In line 174,178, it is stated that the velocity value in the calculation formula of R is based on the ideal velocity rather than the actual observed velocity data. I think it is unreasonable for the statistics of microphysical characteristics. Only the measured value can reflect the real data characteristics. Although there are errors in the measured value, they can be eliminated through necessary quality control methods, which are also done in this study. However, the theoretical value of V is used in the calculation of R in this study. Is there doubt about the quality control method? If so, what is the significance of quality control? In addition, replacing the measured value with the theoretical value makes the feature V unified, which may erase the different characteristics between sites.

Revision: Thank you for the advice. The consideration of theoretical value of V for R was guided by this article Tokay et al. (2014), which illustrated that there would be greater error at the larger end. And in another article Zhang et al. (2019), we saw that the authors used an empirical formula. Combined with the observational environment in this study, there are differences in the subsurface at six sites, and using measured values that have their own errors to calculate would increase the errors in the results. At the time, the results of the two calculation methods were considered in the calculation process and it was found that the differences were relatively small, usually reflected in the second decimal place. Of course, with the advice of the reviewer, we would like to use these rare data for further discussion of the relevant content, including the calculation methods such as the article form Tokay et al. (2014).

Issue 4: There is a conclusion that "the convective events in the Qilian Mountains are more consistent with the continental-like cluster" in lines 367-368. This conclusion is not very convincing. One is all the scatters, and the other is the distribution area of the average values, the two have different meanings and cannot be compared. According to the method of Bringi et al. (2003), the average value should be calculated as well, and compared with the average value region of the two kinds of characteristics obtained in that paper, so that the comparison of the same physical quantity can be more convincing.

Revision: Thank you for your comments. We found that the $D_m$ in the Qilian Mountains was small in both mean or individual samples, which compared with other researches. The results show that the $\log_{10}N_w$ values are not in the range of continental-like cluster or maritime-like cluster, while the $D_m$ values are in the maritime-like cluster. In fact, it could not belong to the maritime-like rain fall. And the results of Bringi et al. (2003) is average value using different samples from different climatic backgrounds, which could be not necessarily comprehensive. In our manuscript, the Qilian Mountains is a special area with unique characteristics of DSD. After considering, there is difficult to say no

way to define whether its precipitation is maritime or terrestrial maritime-like rain fall or continental-like rain fall. Perhaps it can be further discussed in subsequent studies.

Issue 5: The Z-R relationship of DLD is different from that of other sites. Please explain the reasons through specific analysis. The conclusion that "the Z-R relationships of the same section are more consistent" contradicts the unique Z-R relationship of DLD. Why does the Z-R relationship of the TL site with a similar geographical location to DLD differ greatly from that of DLD?

Revision: Thank for the tips. The conclusion that "the Z-R relationships of the same section are more consistent" is seen in conjunction with the previous analysis, where the same section exhibits more similar characteristics such as closer spectral parameters and characteristic variables of DSD. What's more, the distances between the A and b values of any two sites are smallest on the same-side in Figure 11. Based on the above results, it is easily found that the differences in DSD over the southern slopes, northern slopes and interior of the Qilian Mountains are existed with using the data from the eastern and central sites (SD and LB; BLG and HS; DLD and TL) to corroborate each other. However, as the reviewer raised doubt, the uniqueness is still in the specific details, which is mainly due to the fact that each site has its own local climatic influences. In terms of general geographic location, DLD and TL are on the southern slopes of the mountains with in Qinghai Province and at similar elevations. Further analysis, DLD is in Qilian County, which is a narrow valley; while TL is in Menyuan County, which is a relatively open valley. That is to say, DLD is affected by more factors during the rainfall. But if we are in the perspective of the southern slopes of the Qilian Mountains, its uniqueness would not be discussed too much. Of course, as suggested, we have added the relevant content as appropriate.

*other comments:*
Issue 1: "mass-weighted diameters" should be "mass-weighted mean diameters" in lines 20-21..
Revision: Thank for your advice. We have revised the name of parameter.

Issue 2: The statements should be consistent, such as "southern China" in line 62 and "South China" in line 65.
Revision: Thank for your advice. these statements have been consistently expressed as 'southern China' in the text.

Issue 3: Is "Total minutes without noise (min)" in Table 2 not introduced in the text? If not, it is recommended to delete. I think there is something wrong with the expression of "Available rain minutes" in Table 2. It should be a ratio rather than time, and there is no need to add units to the subsequent data.

Revision: Thank for your advice. "Total minutes without noise (min)" is the second corresponding note in the data quality control. "Available rain minutes" is replaced by "Available data", which is relatively straightforward with using the percentage.

Issue 4: In lines 239-240: "It is noteworthy that the frequency of samples with R around 0.6–1.0 mm h$^{-1}$ was highest", this phenomenon is not clearly visible in Figure 3, please mark it in the figure. In addition, the unit in Figure 3 changes after logarithm is taken, it is better to rewrite the unit. For example, $\log_{10}R(mmh^{-1})$ is changed to $\log_{10}R$ (R in $mmh^{-1}$), and $\log_{10}N_w$ and $\log_{10}N_t$ are changed in the same way.

Revision: Thank for your advice. The horizontal coordinate of Figure 3 (e) is "$\log_{10}R$". Corresponding to R in the interval 0.6-1 mm h$-1$, the values of horizontal coordinate roughly are -0.2-0, which is the highest raised part of the curve in the diagram. As Figure 3 consists of 6 small pictures, it has been maintained for the sake of uniformity of the pictures without marked. In addition, the units have been rewritten in Figure 3, including other figures (Figure 5, Figure 7 and Figure 9) with the similar units.

Issue 5: In lines 286-287:" Ma et al. (2019b) also obtained similar conclusions about $D_m$ and $\log_{10}N_w$". What similar conclusions? They should be clearly stated, otherwise there will be misunderstandings. In addition, what is the significance of comparing with other research results, obtaining uniform laws or other?

Revision: Thank for your advice. Similar conclusion from Ma et al. (2019b) is added to the text. "Ma et al. (2019b) also obtained similar conclusions that $D_m$ values increase with the increased rainfall intensity, while the $\log_{10}N_w$ is not as clear." Other research results are cited mainly to set the stage for explaining the variation between raindrop size and number concentration in the follow-up content.

Issue 6: The precipitation type classification is in lines 337-348. It is suggested to add a table to express it more clearly.

Revision: Thank for your advice. We have added flow chart as suggested

[Figure]

Issue 7: lines 363-366: The unit of $N_w$ is"$m^{-3}mm^{-1}$", but that of $log_{10}N_w$ is not.

Revision: Thank for your advice. We have updated this statement in the whole manuscript.

Issue 8: "Black solid lines" and "green lines" are not introduced in the title of Figure 7

Revision: Thank for your advice. We have completed in the title of Figure 7

Issue 9: line 408-409: "the DSD characteristics in the Qilian Mountains consist of a larger $N_w$ and smaller $D_m$" larger or smaller than what?

Revision: Thank for your advice. Compared to the results of studies in other regions, the results of the Qilian Mountains are shown as these characteristics. And we supplemented relevant content in discussion section. As the article covers six sites, it is not convenient to list them directly in the article. There is another article also continuing, which is a selection of one of the sites for analysis, including the relationship between raindrop spectral parameters, and will also compare $N_w$ and $D_m$ in detail with the results of currently available studies.

Issue 10: lines 436-437: "It shows that the Z-R scatter points for HS and BLG were relatively scattered around the 5 mm h$^{-1}$ rain rate." Where is 5mm h$^{-1}$, please mark it in the figure.

Revision: Thank for your advice. It has been marked in the figure.

Issue 11: lines 437-438 "Besides, the Z-R relationship of total rainfall underestimated the stratiform rainfall at low R values and the convective rainfall at high R values", underestimate or overestimate? Please confirm.

Revision: Thank for your advice. This section is mainly based on the results of the fitting in the graph, with reference to Ma et al. (2019b). According to the Z-R fit results, the relationship for total rainfall (black line) has more difference where R is greater compared to the relationship for convective rainfall (red line). And the black line is below the red line, so it is an underestimate. Similarly, the relationship for stratiform rainfall (purple line) is the end where R is smaller.

Issue 12: line 456: What is "SR" short for?

Revision: Thank for your advice. "SR" means "stratiform rainfall". And It has been revised in the article.

Issue 13: lines 492-495: "Compared to previous studies that focused on eastern, southern and northern China as well the Tibetan Plateau, the Qilian Mountains region has its own unique DSD characteristics and Z-R relationship during the rainy season, including a smaller raindrop diameter with a higher number concentration." Please provide a comparison of the specific results in each region, otherwise the conclusion is not convincing.

Revision: Thank for your advice. This question is similar to Issue 9. We have provided a comparison of the specific results in each region, which $D_m$ relates to the raindrop diameter and $N_w$ relates to number concentration. "Compared to previous studies that focused on eastern [3.48 for $\log_{10}N_w$ and 1.23 mm for $D_m$, Pu et al.(2020)], southern [3.86 for $\log_{10}N_w$ and 1.47 mm for $D_m$, Zhang et al.(2019)], northern [3.60 for $\log_{10}N_w$ and 1.15 mm for $D_m$, Ma et al.(2019b)] and central [3.48 for $\log_{10}N_w$ and 1.54 mm for $D_m$, Fu et al.(2020)] China as well the Tibetan Plateau[3.47 for $\log_{10}N_w$ and 1.05 mm for $D_m$, Wang et al.(2021)],"

Issue 14: lines 543-547: "The Z-R relationships in stratiform rainfall were similar and generally underestimated by the $Z=200R^{1.6}$ model used for midlatitude stratiform rainfall; and the Z-R relationships for convective precipitation varied greatly at different stations, which were overestimated by $Z=300R^{1.4}$ at lower rain rates values and underestimated at higher rain rates values." What is underestimated or overestimated, precipitation or the parameters in Z-R relationship? This view is not discussed in the manuscript.

Revision: Thank for your advice. This question is similar to Issue 11. And this view is based on the result given in Figure 11. The Z-R relationship graph is not shown again

because of space issues and it does not look intuitive enough when drawing Z-R relationship graph (such as Figure 10) due to the small differences. Differences of the A and b values are clearer in Figure 11. Here, we have placed the results in the response and not continued to add in the article.

[Figure]

Issue 15: There are some clerical errors or formatting problems in the manuscript. Please check it carefully and make corrections.

Revision: Thank for your advice. We have refined them and checked in the whole manuscript.

---

## Author Response (AR4)

**Response**

Dear Editors and Reviewers:

Thanks for giving us an opportunity to publish our manuscript in this journal. And the comments are very helpful for improving the expression and presentation of the research about raindrop size distribution in Complicated Mountain Terrain.

Please let us know if there is anything need to do during the minor review process.

Authors,

Sincerely.

**Response:**

Reviewer 1

Thanks for the recognition of the manuscript. We will continue to work on the series of researches. And this opportunity makes a lot of sense for us. If the article can be published, it will be the first one after I graduation. Finally, I would like to thank again for the valuable comments, which have made the article better.

Reviewer 2

Thanks for the advice. We have finished minor revision to this manuscript which makes the conclusion more rigorous and adds the more explanation of the site geographical condition. This article means something special to me. And I am so happy it has more improvement through the whole modification process. Finally, I would like to thank again for the reviewer.